# Gingival Necrosis Related to Sepsis-Induced Agranulocytosis Due to *Pseudomonas aeruginosa* Bacteraemia: A Case Report

**DOI:** 10.3390/jcm13051285

**Published:** 2024-02-24

**Authors:** Jia Ying Tan, Guo Nian Teo, Ethan Ng, Andrew Ban Guan Tay, John Rong Hao Tay

**Affiliations:** 1Private Practice, Singapore 428927, Singapore; skyblueaglets@gmail.com; 2Department of Oral and Maxillofacial Surgery, National Dental Centre, Singapore 168938, Singapore; teo.guo.nian@ndcs.com.sg (G.N.T.); andrew.tay.b.g@singhealth.com.sg (A.B.G.T.); 3Department of Restorative Dentistry, National Dental Centre, Singapore 168938, Singapore; etng1324@gmail.com

**Keywords:** sepsis, septic shock, *Pseudomonas aeruginosa*, bacteraemia, agranulocytosis, gingivitis, necrotising ulcerative

## Abstract

**Background**: There have been no reports of sepsis-induced agranulocytosis causing gingival necrosis in otherwise medically healthy patients to the authors’ best knowledge. Even though there are several case reports of gingival necrosis secondary to medication-induced agranulocytosis, they have not systematically described the natural progression of agranulocytosis-related gingival necrosis. **Methods**: This paper presents a case report of a 29-year-old female Indian patient with generalised gingival necrosis and constitutive signs of intermittent fever, nausea, and vomiting. She also complained of abdominal pains. Blood counts showed agranulocytosis, and the patient was admitted for a workup of the underlying cause. Parenteral broad-spectrum antibiotics were administered, which brought about clinical resolution. **Results**: Her gingival necrosis was attributed to sepsis-induced agranulocytosis triggered by *Pseudomonas aeruginosa* bacteraemia, and upon clinical recovery, spontaneous exfoliation left behind exposed bone. Secondary healing over the exposed alveolar bone was noted after a year-long follow-up, albeit with some residual gingival recession. **Conclusions**: Oral manifestations of gingival necrosis, when present with concomitant constitutive symptoms, could indicate a serious underlying systemic condition that could be potentially life-threatening if left untreated. Dentists should be cognizant of this possibility so that timely intervention is not delayed.

## 1. Introduction

Neutrophils are granulocytes that make up 50–70% of the white blood cells in the human body. They mediate inflammatory processes and destroy invading pathogens through phagocytosis, intracellular degradation, granule release, and the formation of neutrophil extracellular traps [1]. Agranulocytosis is a severe form of neutropenia, where the counts of granulocytes are commonly below 0.5 × 10^9^/L [2]. Patients commonly present with fever, malaise, and non-specific pharyngitis, and they may also develop pneumonia and deep infections [3,4]. Common causes of acquired neutropenia or agranulocytosis include malnutrition, infection, bone marrow disorders, autoimmune conditions, hypersplenism, and drugs [3]. Leukocytes, and neutrophils in particular, serve as an important contributor to immunity at the gingival margin [5]. During a neutropenic state, the marginal gingiva is more susceptible to rapidly progressing bacterial challenges that cause cell and connective tissue destruction, especially if mechanical plaque control is inadequate [6,7]. Overgrowth of these commensals can lead to the formation of rampant necrotising gingival lesions and ulcerations in the oral cavity [3,6,7].

*Pseudomonas aeruginosa* is a gram-negative pathogen that is one of the most common causes of bacteraemia, accounting for up to 5.3% of all bloodstream infections [8], and it has the highest mortality rate of about 26–40% [8,9,10,11]. Infection by *Pseudomonas aeruginosa* usually occurs in immunocompromised patients and frequently causes nosocomial infections [11,12]. Such infections can cause necrotic lesions in the oral cavity [13,14,15]. However, there have been instances of severe community-acquired *Pseudomonas aeruginosa* pneumonia resulting in septic shock in immunocompetent patients [16,17,18,19], with resultant neutropenia [20] or even death [16,18,20]. However, oral manifestations are not commonly reported for these cases. Community-acquired pneumonia due to *Pseudomonas aeruginosa* is rare but can be transmitted to medically healthy individuals [20]. It occurs five times more commonly in developing countries as compared to developed countries [21], and predisposing factors include prolonged antibiotic intake [22], poor dwelling conditions [19], and environmental pollution [21,23]. Transmission may occur through the environment, such as through exposure to contaminated water [16,24], and possibly through person-to-person transmission through droplets and aerosols [25,26].

The aim of this present study is to report a novel case of gingival necrosis associated with sepsis in an otherwise medically healthy patient due to a *Pseudomonas aeruginosa* infection, in accordance with CARE guidelines (for CAse REports) (https://www.care-statement.org/, accessed on 14 January 2024).

## 2. Case Presentation

A 29-year-old Indian female presented as an emergency case at the National Dental Centre Singapore in March 2022 complaining of burning pain along her entire gingiva, especially around the upper anterior region. The patient also noticed her gingiva changing colour, and the pain was affecting her sleep for four days. The patient reported no known medical conditions. She was a non-smoker and -drinker and had no history of betel nut consumption or family history of cancer. The patient had not taken any new food or drugs and denied trauma to the area.

The patient reported acute pain along her gingiva around the upper anterior region five days prior and was prescribed Oracort E^®^ (triamcinolone acetonide 0.1%, lidocaine hydrochloride 3%) and Augmentin^®^ (500 mg amoxicillin trihydrate, 125 mg potassium clavulanate) 625 mg twice a day by a medical doctor. The patient noticed that her gingiva, predominantly around the region of her upper right canine to upper left central incisor, developed a purplish patchy appearance, with the central part of the lesion becoming white after two days. She consulted a private dental practitioner for persistent gingival pain and was referred to the Emergency Department of Singapore General Hospital, where she was given intramuscular diclofenac for pain control and referred to our centre for urgent consultation.

On presentation, the patient reported malaise, odynophagia, abdominal pain, and intermittent nausea and vomiting. Her symptoms started roughly a month before presentation since her return from a developing country and were progressively worsening. At triage, she was noted to be febrile and tachycardic. On examination, we noted bilateral submandibular lymphadenopathy and necrosis of the buccal attached gingiva of the upper anterior region, predominantly from her upper right canine to her upper left central incisor, extending up the upper midline frenum (Figure 1).

Mixed purple and white necrotic lesions with sloughing were noted around the buccal gingival margins and attached gingiva of all premolars and molars and palatal/lingual gingival margins of all teeth. More photographs are included in the Appendix A. Probing depths were within normal limits, and no mobility or fetid odour was noted. Radiographic examination revealed normal bone levels and no obvious pathology (Figure 2).

Suspecting an underlying systemic aetiology, the patient was sent for a full blood count, renal panel, liver function test, and human immunodeficiency virus screen, which showed severe leukopenia with 0.19 × 10⁹/L. The numbers for individual types of leukocytes (i.e., neutrophils, lymphocytes, monocytes, eosinophils, basophils) were so low that they could not be quantified (Table 1).

An emergency referral to the Department of Haematology was made, and the patient started to decompensate when she presented at the Haematology clinic, entering a state of shock, with marked hypotension at 64/45 mmHg and tachycardia at 148 beats/min. Fluid resuscitation was initiated, and a transfer to the medical intensive care unit was carried out. Her blood pressure was fluid responsive, and she was weaned off vasopressor support after a day. Blood cultures were taken, and she was empirically started on intravenous piperacillin/tazobactam 4.5 g six hourly and vancomycin 750 mg 12 hourly. Erect chest radiography and computed tomography of the thoracic/abdomen/pelvis regions showed air-space consolidation in the lower lobes of the lungs, which was suggestive of pneumonia (Appendix A). In addition, the blood cultures reported pansensitive *Pseudomonas aeruginosa* bacteraemia, and her human immunodeficiency virus screen was negative. This allowed for a diagnosis of community-acquired *Pseudomonas aeruginosa* pneumonia that led to sepsis and septic shock, resulting in agranulocytosis. No other systemic health conditions such as autoimmune, cardiovascular, or endocrine disease were detected. Vancomycin administration was paused, and the patient improved clinically, consistent with an up-trending white blood cell count over the course of her stay, eventually completing seven days of intravenous piperacillin/tazobactam. She was discharged after a week of inpatient stay and prescribed another week of oral ciprofloxacin, ascorbic acid, and 0.2% chlorhexidine mouthwash and advised for soft diet only.

The patient returned for a dental review a week after discharge on 6 April 2022, with severe pain at her periodontium. Exposed bone was present at the attached gingiva of previously necrotic areas (Figure 3A).

The mucosa around the exposed bone was friable and bled on palpation. No discernible bone loss was noted on radiographic examination.

The patient was advised to avoid brushing her teeth or consuming hard, spicy, or hot food and to continue chlorhexidine mouthwash and analgesics when needed. A week later, the patient no longer had pain but experienced sensitivity due to recession. She was advised to resume toothbrushing and was prescribed vitamin B supplements, iron, and folic acid. Her teeth exhibited normal results on sensibility tests during all reviews, and the sensitivity improved over time. Her oral mucosa continued to heal over the exposed bone over the next eleven months. Professional mechanical plaque control was carried out in July and September 2022 while the patient continued to be on chlorhexidine mouthwash. A localised abscess around the exposed bone around #12 developed at the end of September 2022, which resolved with drainage and saline irrigation. Superficial mobile bony sequestra were noted at the buccal of the upper right lateral incisor and upper right second molar measuring 6 × 8 mm and 7 × 3 mm, respectively (Figure 3B), in December 2022, and they spontaneously exfoliated in February 2023.

By March 2023, all previously affected areas had fully mucosalised, although the buccal attached gingiva of her upper right first premolar to upper left central incisor remained erythematous and spongy, with no associated deep-probing depths (Figure 4A). Buccal recession around her upper central incisors was observed.

Oracort E^®^ (triamcinolone acetonide 0.1%, lidocaine hydrochloride 0.3%) was prescribed, but the patient discontinued use after two days, as she found the texture uncomfortable. The spongy erythematous lesion at the buccal of the upper right canine to upper left central incisor was still present at the 18-month review but had decreased in size (Figure 4B).

Creeping attachment of the previously receded buccal gingiva of the upper right lateral and central incisors was observed despite the remaining presence of black triangles at the area. The lesion was tender to palpation but otherwise asymptomatic. The lesion will be monitored closely at regular reviews. A detailed timeline can be found in Figure 5.

## 3. Discussion

Sepsis is a dysregulated host response to infection that results in life-threatening organ dysfunction. Septic shock occurs when a patient with sepsis also has hypotension that persists despite adequate volume resuscitation, and it requires treatment with vasopressors [27]. Leukopenia can also occur during the immunosuppressive phase of sepsis due to reduction in bone marrow production or increased destruction of granulocytes [28,29]. In this report, the patient experienced agranulocytosis due to septic shock induced by *Pseudomonas aeruginosa* bacteraemic pneumonia and required ionotropic and fluid support. Though extremely rare, we hypothesise that this patient was exposed to contaminated water containing *Pseudomonas aeruginosa*, which resulted in her developing community-acquired pneumonia and subsequent septic shock [20]. As the oral cavity provides a favourable environment for microorganisms to flourish [30], and as local intra-oral bacterial challenges could not be contained due to her neutropenia, this resulted in rapid cell and connective tissue destruction of her marginal gingiva and alveolar bone exposure [6,7]. Bacteria colonisation on the exposed bone may have stimulated bone necrosis [31,32,33] and, combined with possible low-grade trauma to her upper right anterior region, resulted in a localised abscess and bone sequestrum six months later.

*Pseudomonas*-induced sepsis may be associated with dermatologic manifestations including cellulitis, ecthyma gangrenosum, and subcutaneous nodules [34,35] and may also present with lesions in the throat and lips [36]. Further characteristics of agranulocytosis-related gingival necrosis are described in Table 2 and Table 3. Agranulocytosis may be triggered by drugs [6,36,37,38,39,40,41,42], and it commonly occurs in immunocompromised patients [13,14,43,44,45,46,47,48,49,50,51]. Common systemic signs include fever, malaise, nausea, vomiting, lymphadenopathy, pharyngitis, dysphagia, sepsis [37,48], and septic shock [13,36,43], and it may result in death [14,44,45,48].

The patient presented with purple and white necrotic gingiva that exfoliated, leading to bone exposure with eventual localised sequestrum formation. She did not present with any other cutaneous lesions except for the lips. Patients with agranulocytosis may present with pale white necrotic lesions, as a late-stage presentation where the gingiva has already exfoliated [41], or where the necrotic lesions are black or violaceous in colour [43,45]. Oral lesions may also include ulcers or extension of necrosis to the tongue [6,41,48], retromolar pad [48], palate [36,39,40,43,44], floor of the mouth [47,49], lip [39,44], or vestibule [49].

The patient was managed with broad-spectrum antibiotics, and later only with piperacillin/tazobactam. Good oral hygiene was encouraged. Other studies report the use of empirical broad-spectrum antibiotics and/or antifungals [6,36,37,38,39,41,42,44,45,49,50] and substituted them later if cultures were performed [13,14,43,47,51]. Topical treatments include antiseptics [6,13,36,37,38,39,40,46,48,49,50,51], antibiotics [46], or antifungals [39,42,44,46,51] and analgesics [6,42,48] for pain management. Periodontal instrumentation and removal of necrotic tissues, sequestra, and excessively mobile teeth is usually performed when blood counts are deemed safe, and if the zone of necrosis is not too extensive [13,36,37,38,39,40,42,46,49,50]. When possible, drugs that are the suspected cause of decreased blood counts are discontinued or substituted [6,36,37,38,39,40,42,49]. Other treatment modalities include the use of recombinant granulocyte colony-stimulating factor [6,37,43,45,47,49], blood transfusions [40,48,50], policresulen [37], and low-level laser therapy [50]. Complete resolution of gingival healing varies from two days [14] to a year (Table 3).

The spongy appearance of the patient’s anterior maxillary buccal-attached gingiva at the 18-month review is reminiscent of localised spongy gingival hyperplasia, which presents as erythematous, raised areas of attached gingiva [52,53,54,55]. An inflammatory response resulting in hyperplasia has been postulated to be a possible mechanism of pathogenesis [53,56]. The persistent spongy gingival lesion could have been triggered by inflammation due to the extensive necrosis and presence of sequestrum in the area for a prolonged time span. Some cases in the tables above similarly described an irregular granulomatous [36] or a shiny band of erythematous tissue [6,37] that replaced the necrotic gingiva at different time points between 11 days and eight months. Limitations of this report include the unknown source of the patient’s *Pseudomonas aeruginosa* pneumonia. However, the strength of this report is in the novelty of the presentation of gingival necrosis as an early sign of sepsis related to a *Pseudomonas aeruginosa* infection in an otherwise medically healthy patient. Regular reviews are planned for this patient.

## 4. Conclusions

Dentists should be aware and maintain a low threshold for referring patients for a workup and possibly emergent medical management when the oral manifestations of gingival necrosis and ulcerations are accompanied by constitutive changes. Patients that appear toxic should be screened for systemic changes and referred immediately to a tertiary medical care setting for further management.

Emergent care involves assessing the patient for hemodynamic stability and airway security. Close cardiorespiratory monitoring should be initiated even if patients still present as stable on admission, as deterioration in sepsis occurs very quickly. Septic patients often require monitoring and care in an intensive care unit for hemodynamic and ventilatory support. Baseline bloods to establish clinical severity and guide treatment should be obtained as soon as clinically possible, and empiric broad-spectrum antibiotic therapy started. Once emergent care is provided and the patient’s critical condition has been stabilised, further workup to identify the source of infection should be completed, and the infection treated. After medical care has been stepped down, the dentist should help alongside the medical team in providing symptom relief. The patient should be kept on close weekly follow-up with the dentist. The use of topical antiseptics and systemic analgesics should be prescribed, and professional mechanical plaque control should be performed, until all necrotic or ulcerated oral lesions resolve. If there was exposed bone at initial presentation, the site may take an extended period of up to a year to fully heal and may also result in a secondary infection and bone sequestrum, which must be managed with the removal of necrotic tissues and sequestra. Appropriate analgesics and antibiotics should also be prescribed with consultation with the patient’s medical care team as well. The interval between follow-ups can be progressively increased with clinical improvement. The patient should be reviewed until complete resolution is observed.

## Figures and Tables

**Figure 1 jcm-13-01285-f001:**
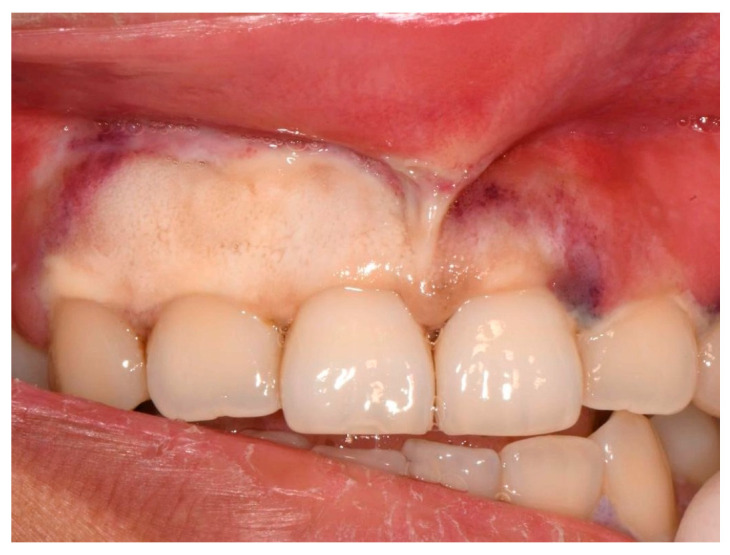
Initial clinical presentation of necrotic buccal attached gingiva (including frenum).

**Figure 2 jcm-13-01285-f002:**
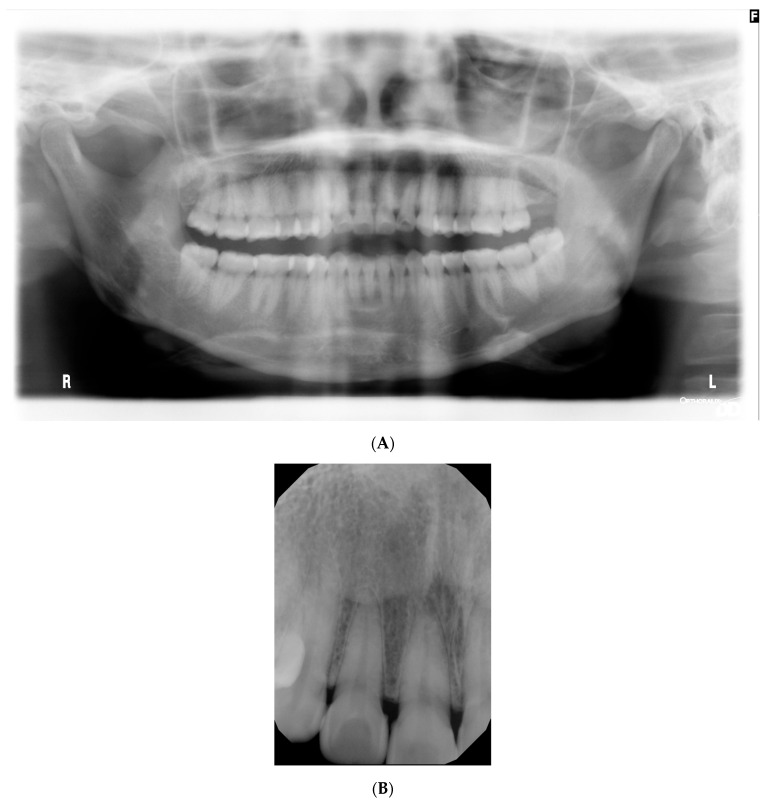
(**A**) Dental pantomogram taken at initial presentation. (**B**) Periapical radiograph taken at tooth #11, #12 mild calculus deposits but no radiographic bone loss.

**Figure 3 jcm-13-01285-f003:**
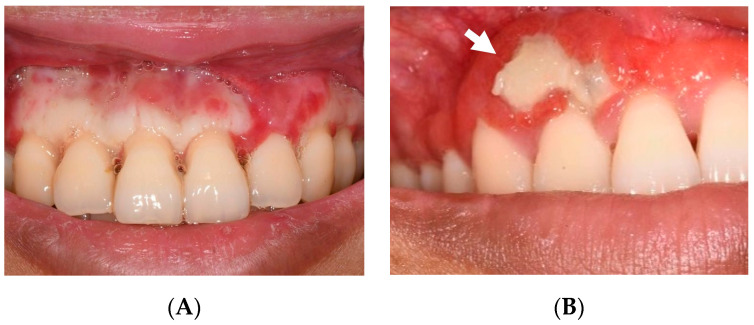
(**A**) Two weeks after initial presentation. Exposed bone at the upper anteriors after the necrotic gingiva sloughed off. (**B**) At nine-month follow-up. Slightly mobile sequestrum (white arrow) at the upper right lateral incisor, which spontaneously exfoliated.

**Figure 4 jcm-13-01285-f004:**
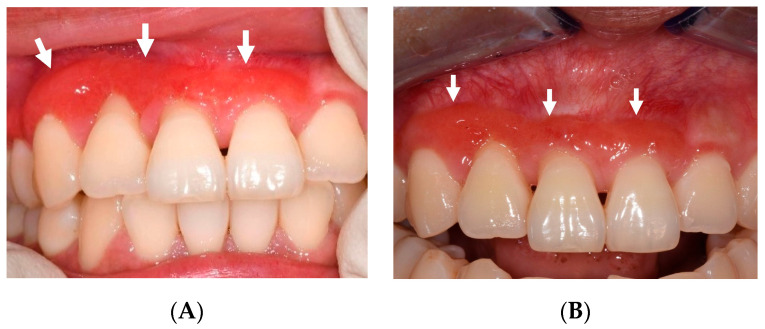
(**A**) Twelve-month follow-up showing erythematous and spongy buccal gingiva (white arrows). Buccal gingival recession is evident. (**B**) Eighteen-month follow-up showing reduction in size of spongy buccal gingiva (white arrows).

**Figure 5 jcm-13-01285-f005:**
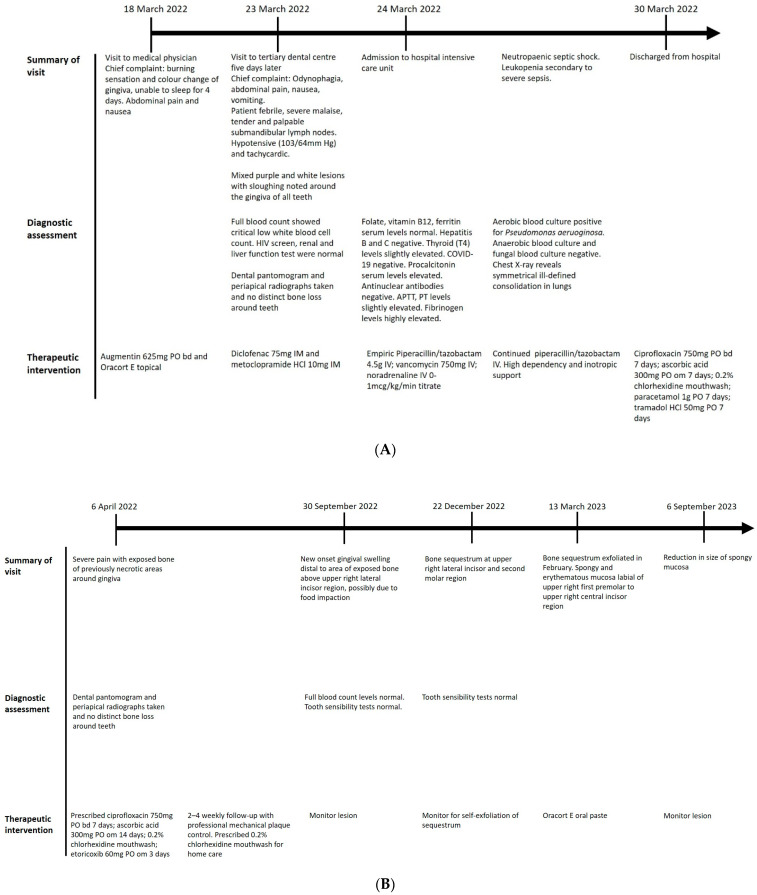
(**A**) Timeline of events. (**B**) Timeline of events (continued).

**Table 1 jcm-13-01285-t001:** Haematological laboratory values taken at initial presentation. No available readings for leukocyte subtypes. Abbreviations: dL, decilitre; fL, femtolitre; MCH, mean corpuscular haemoglobin; MCHC, mean corpuscular haemoglobin concentration; MCV, mean corpuscular volume; pg, picogram; RBC, red blood cell; WBC, white blood cell. ↓, low count; ↓↓, critically low count.

Haemoglobin (g/dL)	WBC Count (×10^9^/L)	Platelet Count (×10^9^/L)	RBC Count (×10^12^/L)	MCV (fL)	MCH (pg)	MCHC (g/dL)	RBC Distribution Width (%)	Mean Platelet Volume (fL)	Haematocrit (%)
↓ 10.8	↓↓ 0.19	182	↓ 4.02	80.6	↓ 26.9	33.3	13.4	10.4	↓ 32.4

**Table 2 jcm-13-01285-t002:** Features of agranulocytosis-related gingival necrosis and their presentation.

Reference	Cause of Agranulocytosis	Patient Characteristics	Systemic Signs	Extra-Oral Lesions	Septic Shock	Extent of Necrosis	Other Intraoral Lesions	Bone Exposure
Swenson et al., 1965 [41]	Suspected to be due to medication: Alka-Seltzer, aspirin, Haley’s M-O, penicillin, antihistamine, Tofranil, Milpath	59-year-old white female, non-smoker, non-drinker, no history of narcotic use	Severe heartburn and indigestion two months before. Hepatitis, jaundice, malaise, pharyngitis, nausea, vomiting	NA	NA	Absence of almost all marginal and attached gingiva in maxilla and mandible, buccally and lingually. Only exposed bone noted	1 cm gangrenous ulcer on tongue	Y, already formed sequestration at first presentation. Exposure depths 1–4 mm. 2–3 mm of sequestrum separated from alveolar bone on X-ray
Hou and Tsai, 1988 [36]	Methimazole	34-year-old female with hyperthyroidism	Pharyngitis, fever, chills, insomnia for two months. Sudden-onset pneumonia thereafter	Tonsillar enlargement with pus coating in the crypt. Red swollen painful wound on left index finger. Yellow crust on lips	Y	Generalised gingival necrosis with numerous diffuse ulcerative lesions	Spread to hard and soft palate	NA
Ohishi et al., 1988 [42]	Suspected to be due to medication: sulpyrine, cefmetazole, indomethacin	47-year-old female with springtime pollinosis	Fever, chills, nausea, vomiting, loss of consciousness	NA	NA	Generalised, involving all free and attached gingiva	Heavy coating on tongue, white plaques on palate	NA
Myoken et al., 1995 [44]	AML on chemotherapy (etoposide, cytarabine, epirubicin, vindesine, mercaptopurine, prednisolone)	73-year-old male with AML	Fever, dysphagia, hypersalivation	None	NA	Not clearly mentioned but at least upper anteriors	Spread to upper lip and palate	Y, necrotic
Myoken et al., 1999 [43]	AML on chemotherapy	48-year-old female with AML	Fever	None	Y (Sepsis with hypotension)	Palatal gingiva of two teeth †	Small portion of palate close to the two teeth	NA
Myoken et al., 2002 [45]	Diffuse large B-cell lymphoma, non-Hodgkin lymphoma starting chemotherapy (doxorubicin, cyclophosphamide, vindesine, prednisolone)	66-year-old Japanese female with diffuse large B-cell lymphoma, non-Hodgkin lymphoma	Fever, hypersalivation	Lung nodules	NA	Left maxilla †		Sequestrum present within gingiva
Barasch et al., 2003 [14]	ALL with intraluminal catheter placement the day before (no chemotherapy started)	23-year-old female with ALL	Fever, pain at catheter site	NA	NA	2.5 cm necrosis at anterior left maxillary gingiva	NA	NA
Tewari et al., 2009 [40]	Long-term deferiprone	14-year-old male with β-thalassemia major, on regular blood transfusions every two to three weeks	High-grade fever, malaise, and generalised weakness, loss of appetite, lymphadenopathy, and dysphagia. Extreme pallor, bilateral submandibular lymphadenopathy, and splenomegaly	Necrosis around lip commissures when medical intervention started	NA	Around all surfaces of all teeth	Left palate to median palatal raphe, spreading to right palate when medical intervention started	Y
Kim et al., 2015 [37]	Methimazole	31-year-old female with hyperthyroidism	Pharyngitis, fever	NA	Unclear. Steroids administered to “control severe inflammatory reactions”	Generalised from free gingival margin to mucogingival junction, also involving palatal and lingual aspects	NA	NA
Xing and Guan, 2015 [39]	Propylthiouracil	43-year-old Chinese female with hyperthyroidism	Dysphagia, reduced appetite	NA	NA	Extensive, at least around incisors and premolars	Started as inner lip ulcer that spread to gingiva. Hard palate necrosis	Y, partially necrotic
Amirisetty et al., 2016 [46]	Stage I CLL on chemotherapy (bendamustine, rituximab) and IV antibiotics three days prior due to skin lesions (unknown antibiotic)	51-year-old male with stage I CLL	Palpable lymph nodes, fever, vomiting	Maculopapular rashes on forearms and palms	NA	Buccal and lingual aspects of tooth #22–#24 25 mm × 8 mm	NA	Y
Chang et al., 2017 [6]	Methimazole	25-year-old Asian female with Grave’s disease	Fever, pharyngitis, oral ulcer, chills, dysphagia, trismus	Pharyngeal ulcers	NA	Generalised	Tongue ulcers	NA
Arora et al., 2018 [38]	Methimazole	46-year-old female with hyperthyroidism	Sore throat, fever associated with chills, swelling over nose, gingival pain, trismus	Left forearm ~2 cm black necrotic ulcer, multiple ~1 cm pustular nodular erythematous lesions on her nose and back	NA	Buccal gingiva of #14, #15, #17, #25, #26, #33, #34, #35, #42, #44, and #45	Angular cheilitis	NA
Boddu et al., 2018 [47]	AML on chemotherapy (cladribine, idarubicin, decitabine)	62-year-old female with myelodysplastic syndrome that transformed into AML	NA	Later developed skin lesions on forehead and shoulders, multiple nodular pulmonary lesions	NA	Buccal of incisors and premaxillary region nearest the interincisal papillae. Gingival hyperplasia also noted	Sublingual ulcers	NA
Fatahzadeh, 2018 [49]	Kidney transplant, end-stage renal disease, diabetes mellitus (amongst other conditions), on mycophenolic acid, tacrolimus, valganciclovir, etc.	58-year-old male with multiple comorbidities. Previous surgeries include kidney transplantation (2016) and dialysis fistula surgery	NA	NA	NA	Lower incisors, palatal of #18	Necrosis extended to anterior floor of mouth and buccal vestibule	NA
Jandial et al., 2018 [51]	AML, post-allogenic hematopoietic stem cell transplant with graft failure	19-year-old female with AML, post-allogenic hematopoietic stem cell transplant with graft failure	Fever, tachycardia, tachypnoea, trismus, pain on chewing	NA	NA	Generalised gingival erythema and oedema. 10 × 10 mm necrotic plaque along right upper premolar	NA	NA
Souza et al., 2018 [13]	HIV +ve, chronic kidney disease, *P. aeruginosa* pneumonia	6-year-old female with chronic kidney disease undergoing peritoneal dialysis, HIV +ve	Fever, difficulty eating and drinking	NA	Y	Around all surfaces of all teeth	N	Y, osteomyelitis
Boras et al., 2019 [48]	AML and dilated cardiomyopathy on meds. On chemotherapy (azacytidine)	40-year-old female in remission for stage IVA diffuse large B-cell lymphoma (10 months ago), with dilated cardiomyopathy and AML	NA	NA	*Staphylococcus hominis* sepsis (unknown if shock present)	1 cm diameter lesions on vestibular mucosa above #21, #22, #25, and #26, and around #45 and #46	Ulcers on tongue and retromolar area	NA
Ximenes et al., 2021 [50]	Paroxysmal nocturnal haemoglobinuria associated with aplastic anaemia	60-year-old male with paroxysmal nocturnal haemoglobinuria associated with aplastic anaemia, defaulted eculizumab (for his condition) for 45 days	NA	NA	NA	2 cm necrosis palatal to #24	Fistula at #24 buccal	NA

† Necrotic tissue had a violaceous/black appearance instead of whitish yellow. Key: NA, not mentioned in paper; Y, yes; +ve, positive. Abbreviations: AML, acute myelogenous leukaemia; ALL, acute lymphoblastic leukaemia; CLL, chronic lymphocytic leukaemia; HIV, human immunodeficiency virus; IV, intravenous.

**Table 3 jcm-13-01285-t003:** Management and outcomes of agranulocytosis-related gingival necrosis.

Reference	Relevant Swab/Histology/Culture Results ††	Dental Treatment Protocol	Medical Treatment	Time to Resolution	Dental Complications
Swenson et al., 1965 [41]	NA	Scaling performed. Gingiva improved but bone remained. Sequestrum all removed but teeth remained excessively mobile, necessitating full mouth clearance after three months	Administered penicillin, erythromycin, prednisolone, Surbex	Within three months	Generalised severe mobility and self-exfoliation of one tooth. Likely had pre-existing periodontitis
Hou and Tsai, 1988 [36]	Throat +ve *Pseudomonas aeruginosa* pneumonia Blood −ve	2.5–3% H_2_O_2_ and 2% CHX for irrigation twice weekly 0.2% CHX rinse while admitted Scaling and root planing, and OHI when gingiva improved after two weeks	Discontinued methimazole Administered amikacin, piperacillin	Shedding from day 11 Resolution around three months	Recession
Ohishi et al., 1988 [42]	Swab +ve fungus (species not mentioned)	Lidocaine viscous, amphotericin B syrup (10× dilution) Scaling after symptoms subsided	Discontinued all medication except antibiotics and steroids Given vitamin B2, B6, B12	Shedding from day 13 Resolution on day 20	Recession
Myoken et al., 1995 [44]	Histology and swab +ve for *Fusarium moniliforme* Blood −ve (already on antibiotics and antifungals)	Oral rinses with amphotericin B syrup (l0 mg/mL)	Before lesions: laminar airflow room, administered broad-spectrum antibiotics, empirical fluconazole, and amphotericin B syrup Fluconazole changed to amphotericin B on alternate days when granulocytopenia noted	NA (patient death)	NA (patient death)
Myoken et al., 1999 [43]	Blood, sputum, histology +ve for *Pseudomonas aeruginosa*	NA	Before lesions: laminar airflow room When ANC = 0, administered empirical panipenem, amikacin, fluconazole, itraconazole, and recombinant granulocyte-colony stimulating factor. Later, medications changed to meropenem, amikacin, then oral levofloxacin	25 days	NA
Myoken et al., 2002 [45]	Blood and urine −ve, histology +ve and culture +ve for *Trichoderma Longibrachiatum*	NA	Before lesions: laminar airflow room, prophylactic itraconazole, amphotericin B, tosufloxacin Administered amikacin, panipenem/betamipron, amphotericin B, recombinant human granulocyte colony-stimulating factor (started before necrosis). Later continued amphotericin B	NA (progressively worsened)	NA (patient death)
Barasch et al., 2003 [14]	Blood −ve but elevated HSV antibodies Swab +ve HSV and *Pseudomonas aeruginosa*	NA	Administered empirical ceftazidime, acyclovir (added when HSV antibodies were elevated)	Two days	NA (patient death)
Tewari et al., 2009 [40]	Blood −ve and swab −ve *Staphylococcus* spp., *Fusobacterium* spp., *Treponema* spp. noted in crushed necrotic tissue	Supragingival debridement without disturbing involved tissues and profuse subgingival irrigation with 1% povidone iodine and H_2_O_2_ Necrotic tissues and loose sequestrum removed without anaesthetic when they started detaching Hawley retainer issued to protect against sensitivity when eating 0.12% CHX rinse	Discontinued deferiprone transfusions Administered cefepime, metronidazole, linospan, and ornidazole	Shedding at day 10 Resolution at two months	Progressed to acute periodontitis with some mobility around the maxillary and mandibular canines and premolars, but without interdental crater formation Sequestrum at 14 days
Kim et al., 2015 [37]	NA	50% policresulen and 0.1% CHX, then policresulen substituted for H_2_O_2_ when shedding started When WBC stabilised, periodontal debridement performed during every follow-up visit	Discontinued methimazole Administered amoxicillin/clavulanic acid, isepamicin sulphate, granulocyte colony-stimulating factor To control inflammation, administered dexamethasone disodium phosphate and later hydrocortisone sodium succinate	Shedding at day three Gingiva grew back within two weeks, but complete healing noted at one year	Sequestrum at 21 months Black triangles (mild)
Xing and Guan, 2015 [39]	Culture +ve *Candida albicans* (unclear sampling site) Refused bacterial culture Non-specific histologic results for microbiology	3% H_2_O_2_ supragingival irrigation, ultrasonic scaling CHX, nystatin, OHI Planned for regular supragingival ultrasonic scaling, surgical removal of necrotic bone (when thyroid function was normal, radioactive iodine treatment was recommended), extractions of periodontally involved teeth, soft tissue grafting when she returned	Discontinued propylthiouracil Administered metronidazole and amoxicillin	Regeneration around 10 months, except around the surface of the necrotic alveolar bone. Patient had rejected treatment earlier due to cost	Osteonecrosis Severe mobility of mandibular incisors, moderate mobility of upper incisors and posteriors (splinted bridge) Likely had pre-existing periodontitis Recession after small necrotic bone particles were removed
Amirisetty et al., 2016 [46]	Histology of necrotic tissue showed *Candidal* hyphae	Necrotic tissue removed after confirming the absence of thrombocytopenia. Site was cleaned w 3% H_2_O_2_, SRP with hand instruments performed Metronidazole 1%/CHX 0.25% gel empirically prescribed. Changed to clotrimazole cream after final diagnosis	Prescribed topical vitamin K for skin lesions	Skin lesions resolved at one month Gingival lesions resolved at three months	Unclear if recession fully resolved
Chang et al., 2017 [6]	Blood −ve	0.2% CHX and 0.05% lidocaine, encouraged brushing while admitted	Was on amoxicillin, sulfamethoxazole, acyclovir before admission, but did not stop methimazole Discontinued methimazole on admission Ceftazidime swapped for ciprofloxacin (due to mildly elevated eosinophil count), recombinant granulocyte colony-stimulating factor	More than eight months (still erythematous and had sequestrum around two teeth)	Sequestrum spontaneously exfoliated at eight weeks, resulting in reduction in pain
Arora et al., 2018 [38]	NA	H_2_O_2_ and CHX while admitted Improvement after one week Periodontal debridement and 0.2% CHX at discharge	Discontinued methimazole Administered IV antibiotics	Around six months	NIL
Boddu et al., 2018 [47]	Blood −ve Skin histology and oral histology +ve *Fusarium*	Oral rinses (unknown)	Administered empirical broad-spectrum antimicrobials, caspofungin, posaconazole, keratinocyte growth factor and recombinant granulocyte colony-stimulating factor Added liposomal amphotericin, voriconazole, posaconazole, and WBC transfusions after fungal diagnosis	Improvement noted, but patient died due to other causes	Suspected osteomyelitis on scans
Fatahzadeh, 2018 [49]	Histology +ve gram +ve bacteria including *Actinomyces*	Removal of involved mobile teeth and debridement of necrotic areas under antibiotic prophylaxis with amoxicillin, post-op amoxicillin, and CHX 0.12% Scaling and polishing when more stable	Discontinued mycophenolic acid Administered recombinant granulocyte colony-stimulating factor	Not mentioned, but resolved	Severe bone loss and mobility at lower incisors Likely had pre-existing periodontitis
Jandial et al., 2018 [51]	Blood and urine −ve Swab +ve for *Pseudomonas aeruginosa*	1% clotrimazole mouth paint and CHX	Administered empirical cefoperazone, sulbactam, vancomycin with supportive measures (fentanyl transdermal patch and morphine IV boluses for pain relief) Due to sterile blood culture report and continuous high-grade fever, antibiotics were changed to cefepime and amikacin	Shedding completed by day 17	NA
Souza et al., 2018 [13]	Blood and nasal +ve, gingival swab and bone culture +ve for *Pseudomonas aeruginosa*	Supragingival scraping 0.12% CHX rinse Bone lesions and mobile primary teeth surgically removed under antibiotic prophylaxis, with one week of post-op antibiotics	Administered vancomycin, ampicillin-sulbactam, amikacin, piperacillin-tazobactam, and polymyxin B, then levofloxacin for 15 days	15 days after surgery	Mobility, bone loss, and CAL without pocketing around central incisors and first molars Chronic hematogenous osteomyelitis
Boras et al., 2019 [48]	Fungal/viral swab −ve, later *Stenotrophomonas maltophilia* and *Enterobacterium faecalis* +ve, but suspect concomitant finding as ulcers were there for three months before bacteria were found	CHX, lidocaine gel, Gelclair^®^ (Helsinn Healthcare SA, Lugano, Switzerland), benzydamine hydrochloride, betamethasone in Orabase^®^	Blood transfusions	NA (patient death)	NA (patient death)
Ximenes et al., 2021 [50]	Biopsy No swab or microbiological examination performed	Minimally traumatic excisional biopsy and extraction of associated tooth with local haemostatic measures and pre-op antibiotics despite no improvement in blood picture Low-level laser therapy CHX	Administered Augmentin^®^ Blood transfusion when blood picture did not improve	At least 30 days	NA

†† Negative bacterial/fungal/bacterial swabs and other tests were left out unless relevant. Blood cultures included to illustrate presence or absence of bacteraemia. Key: NA, not mentioned in paper; +ve, positive; −ve, negative. Abbreviations: ANC, absolute neutrophil count; biopsy, bx; CAL, clinical attachment loss; CHX, chlorhexidine; H_2_O_2_, hydrogen peroxide; HSV, herpes simplex virus; IV, intravenous; OHI, oral hygiene instructions; WBC, white blood cell.

## Data Availability

All data generated or analysed during this study are included in this published article and its Appendix A.

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
