# Peer review of "Gingival Necrosis Related to Sepsis-Induced Agranulocytosis Due to Pseudomonas aeruginosa Bacteraemia: A Case Report"

_jcm, 2024, doi:10.3390/jcm13051285_

Round 1

Reviewer 1 Report

Comments and Suggestions for Authors

After reviewing the manuscript with identification number jcm-2850117, we have the following comments. 

In general, the document is well written, the information and the blocks of information are handled in a pertinent and adequate manner, the guidelines for the presentation of clinical cases are followed. 

In the title:

It is suggested to change the word "associated" to "related", since there is only one case presentation, and it would be very adventurous to indicate a plausible and causal association of gingival necrosis with agranulopathy.

In the abstrac:

In line 16, it is mentioned that 21 articles were reviewed, here the question arises as to why 21, what were the criteria for inclusion and exclusion, elimination. To avoid this type of questioning at such an early reading of the manuscript, it is suggested to change to another sentence that denotes only the revision and updating of concepts from the available literature; another point would be to develop the materials and methods of the bibliographic review in its narrative or systematic modality. 

In the introduction:

In the line 60, it is suggested to homologate this information, first determining the general and specific objectives, and then being constant and similar between the information sections. 

In the case presentation:

It is suggested to specify the anatomical area affected, not only mentioning "gingival mucosa", it is more convenient and adequate to place "in upper right quadrant/ Quadrant 1/ area that contemplates the gingival mucosa from dental organ 1.3 to 2.1... for example. Be specific in the area of pain, discomfort, anatomical changes, signs and symptoms where the alteration occurs. Check Lines 65 and 70.

In line 71, change for Lidocaine. 

Lines 78,79:

Specify whether the patient was in contact with communities or populations known to have this disease, so that it can be called "Community-acquired pneumonia due to Pseudomonas aeruginosa".

In line 80:

It would be very informative to know if the patient noticed any color change prior to the examination performed. The presentation of the case is important, but the previous data collected up to the initial day of examination are a fundamental part in order to recognize and understand the pathophysiology of this condition.

In ljne 90:

In the panoramic radiograph, it is difficult to identify the loss of bone crests, the most appropriate would have been to take periapical radiographs or bitewing to indicate whether or not there was a bone loss around the teeth involved by the involvement of the gingival mucosa. Improve the wording of this statement.

Line 154, change for lidocaine.

it is suggested, to improve the robustness of the document in its scientific structure, to place a small section that denotes the materials and methods of the literature review; it can be as simple as "For the literature review, we chose to make a narrative review taking into account the available literature on the topic of interest, through search engines such as google academic, databases such as PUBMED, NCBI, Scopus, Elsevier. ... in any language, from such day to such day, freely accessible or accessible via free of charge and/or subscription, and containing relevant information regarding the features, management, and outcome of this disease, for further analysis and interpretation. 

Grateful for the opportunity to be part of the peer review committee, reaffirming my commitment to such task with professionalism, ethics and responsibility. 

Best Regards 

Author Response

Please see the attachment with our responses in red.

Reviewer 2 Report

Comments and Suggestions for Authors

First, I would like to thank the opportunity of reviewing for JCM.

The manuscript refers to a case report associated to a literature review concerning Gingival necrosis associated with sepsis-induced agranulocytosis due to Pseudomonas aeruginosa bacteraemia.

Overall the subject is of JCM's readers interest. 

Authors are encouraged to correct some items:

1. All keywords should be corrected to strictly meet MeSH terms: as an Health Science's journal, the use of MeSH terms can avoid scientific's future search errors.

2. Entire text proofreading is highly recommended. Example: LINE 13 - "... otherwise medic ally healthy patients to the authors’..."  (medically does not have this space...)

3. REFERENCES: using recent references (5 years or less can guarantee the subject is updated. Please review cited references and include more recent literature.

4. Figures: use of arrows and others makers is highly recommended to show readers what is important to focus.

Comments on the Quality of English Language

Entire text proofreading is highly recommended. Example: LINE 13 - "... otherwise medic ally healthy patients to the authors’..."  (medically does not have this space...).

Author Response

Please see the attachment with our replies in red.

Reviewer 3 Report

Comments and Suggestions for Authors

It was evaluated the article titled “Gingival necrosis associated with sepsis-induced agranulocytosis due to Pseudomonas aeruginosa bacteraemia: A case report and literature review”.
I do not recommend to use two types of articles in one (Case report and Review).
For the table presented, e.g., there are biases involved.

The goal of this case report was to present “a novel case of gingival necrosis associated with sepsis in an otherwise medically healthy patient due to a Pseudomonas aeruginosa infection”. I did not consider it an innovation; I considered it a misunderstanding or misinterpretation. I do not recommend to include this affirmation in the title.

- Lines 110-111: Include image of the CBCT cited
- where is the results of the blood culture? Only P. aeruginosas was found?
- where are the periodontal evaluation of the patient?

I considered the presentation of the case a little bit weird. I strongly recommend to improve the description and better presenting the findings.

Author Response

Please see the attachment and our responses in red.

Reviewer 4 Report

Comments and Suggestions for Authors

The case and topic choosed for this manuscript are really interesting and important for the clinical medicine. Description of the signs and symptomes produced by the patient, together with the illustrations (figures, tables) is detailed showing impressive picture about the problems. In my opinion it was a good idea to present the experiences supporting the importance of such relatively rare infection caused by Pseudomonas aeruginosa. Pseudomonas induced sepsis may be associated with different systemic and local symptomes that could be resulted in serious consequences. So, this condition needs interdisciplinary treatments.

Supplementary materials represent a good summary of the related information and literature.

The manuscript is a useful special material which calls the dentists’ attention for the possibilities of serious situations due to Pseudomonas infections, furtherly for the importance of proper diagnosis performed in time and for the necessity of good collaborations between the specialists of different medical disciplines.

In conclusion the content of the manuscript has great importance for the every-day practice but the whole manuscript would need language corrections.

Comments on the Quality of English Language

Some corrections editing are needed.

Author Response

Please see the attachment and our replies in red.

Reviewer 5 Report

Comments and Suggestions for Authors

The manuscript aims to present a clinical case of gingival necrosis associated with sepsis-induced agranulocytes and a literature review. Although the case report is well presented and discusses symptoms and diagnosis, the manuscript does not constitute a literature review.

Table 2 presents the different agranulocyte-induced gingival necrosis citations; the authors do not discuss or present them as a literature review.

The manuscript's discussion improved with the addition of these citations, but there needs to be an actual review to consider the present manuscript as a review.

However, the case report is interesting, and the authors have good points in the manuscript. My primary suggestion would be for the authors to remove "and literature review" from the title.

The additional few points below require attention.

The abstract should include the cause of the sepsis-induced agranulocytosis. The information is in the title but must be in the abstract. That would help the readers engage with the manuscript's flow.

Table 2 would be improved by adding more information about the cases cited in the articles. For example, age range, gender, clinical groups (healthy, not healthy, diabetes, etc.)

Due to the impact of diabetes on the granulocyte population and oral and systemic health, I am just curious if they screened patients for diabetes at any time.

At any time in the clinic, they collect samples from the site and perform microbiological detection. It would be interesting to know what bacterial populations may be present besides the cell infiltrate in that lesion.

Author Response

Please see the attachment and our reponses in red.

Round 2

Reviewer 2 Report

Comments and Suggestions for Authors

Authors have adapted text to meet first revision suggestions.

Author Response

We thank the referee for the kind comments and thorough review of the manuscript.